# EDGE-GRPO: Entropy-Driven GRPO with Guided Error Correction for Advantage Diversity

## Abstract

Large Language Models (LLMs) have made remarkable progress in enhancing step-by-step reasoning through reinforcement learning. However, the Group Relative Policy Optimization (GRPO) algorithm, which relies on sparse reward rules, often encounters the issue of identical rewards within groups, leading to the advantage collapse problem. Existing works typically address this challenge from two perspectives: enforcing model reflection to enhance response diversity, and introducing internal feedback to augment the training signal (advantage). In this work, we begin by analyzing the limitations of model reflection and investigating the policy entropy of responses at the fine-grained sample level. Based on our experimental findings, we propose the EDGE-GRPO algorithm, which adopts **E**ntropy-**D**riven Advantage and **G**uided **E**rror Correction to effectively mitigate the problem of advantage collapse. Extensive experiments on different models across multiple main reasoning benchmarks demonstrate the effectiveness and superiority of our approach. The code and weights will be released upon acceptance to facilitate further research in this field.

## 1 Introduction

Recent advancements in large reasoning models, such as OpenAI-o1 Jaech et al. (2024) and Kimi-K1.5 Team et al. (2025), have shown impressive progress in complex tasks involving mathematics and coding. Among them, the Group Relative Policy Optimization (GRPO) algorithm Shao et al. (2024) has attracted considerable attention from researchers. By discarding the value function used in the PPO algorithm Schulman et al. (2017) and instead computing rewards and relative advantages across sampled responses within each group, it significantly reduces resource consumption during training while improving reasoning performance.

For the computation of rewards in the GRPO algorithm, some studies adopt a Process Reward Model (PRM) to provide more fine-grained feedback Cui et al. (2025); Wang et al. (2025b). However, it introduces substantial computational overhead. As a result, other works abandon the

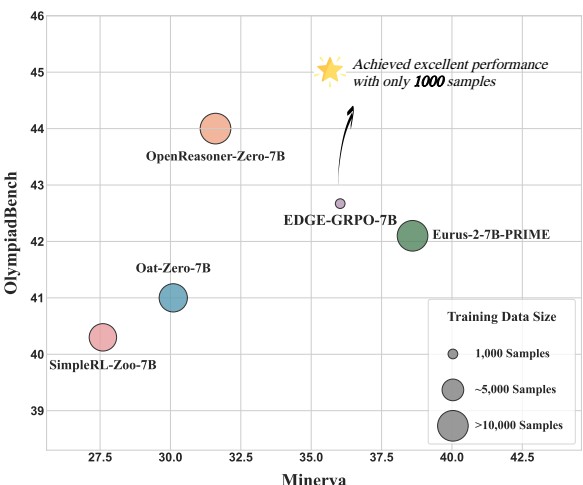

Figure 1: Performance comparison with other open-source models on Olympiad and Minerva. Our method achieves competitive and excellent performance with only 1K training samples. These models are all post-trained based on Qwen2.5-Math-7B.

reward model in favor of using rule-based reward functions Zhou et al. (2025); Zhang et al. (2025b). However, this often leads to sparse rewards, where all responses within a group receive identical rewards. Consequently, the calculated advantages for each response become zero, cease to provide

effective policy gradient, and lead to advantage collapse during training. This phenomenon severely limits the efficiency of the sample.

Recent studies have primarily sought to alleviate this problem from two perspectives. At the response level, efforts focus on increasing the diversity of responses to prevent identical rewards across all responses, such as enforcing model reflection on incorrect answers to reduce the occurrence of uniformly incorrect outputs within a group Wang et al. (2025a); Wan et al. (2025). However, the extent to which reflection contributes to performance improvement remains inconclusive. At the signal level, internal feedback is introduced to augment the advantage, such as incorporating response-related semantic entropy or policy entropy into the advantage calculation Chen et al. (2025); Cheng et al. (2025). However, most studies either pursue low entropy to improve accuracy or encourage high entropy to maintain exploration, lacking fine-grained modeling of the relationship between responses and their policy entropy.

In this paper, we begin by analyzing the limitations of model reflection. Quantitative experiments show that responses containing self-reflection are often associated with significantly lower accuracy. Although forced reflection can help the model correct a subset of answers, its overall effectiveness remains limited. Additionally, we observe a misconception in the model's estimation of policy entropy at the fine-grained sample level: incorrect responses do not necessarily indicate uncertainty, some of them exhibit notably lower entropy. Conversely, the model is not always confident in its correct responses, some of which display relatively high entropy.

To address these issues, we propose a simple and effective EDGE-GRPO (**E**ntropy-**D**riven GRPO with **G**uided **E**rror Correction) algorithm. At the response level, we introduce Guided Error Correction (GEC) to enhance response diversity, providing more effective guidance even when the model encounters questions beyond its current capacity. At the signal level, we compute an Entropy-Driven Advantage (EDA) that assigns higher advantages to correct responses with low entropy and lower advantages to incorrect responses with low entropy, thereby increasing the diversity of the advantage signal. These improvements significantly mitigate the problem of advantage collapse. Across multiple reasoning benchmarks, our method achieves substantial performance gains compared to the vanilla GRPO. As shown in Figure 1, our approach reaches comparable performance to other open-source models using only 1K training samples.

Our contributions can be summarized as follows:

- We analyze the key challenges faced by preliminary attempts. Specifically, at the response level, prompting the model to reflect on incorrect responses has limited effectiveness. At the signal level, fine-grained sample-level policy entropy is needed to guide the augmentation of the advantage.

- We propose the EDGE-GRPO algorithm. At the response level, we introduce Guided Error Correction (GEC) to overcome the limitations of the model capacity and improve response diversity. At the signal level, we compute an Entropy-Driven Advantage (EDA) to increase the diversity of the advantage signal, significantly alleviating the problem of advantage collapse.

- Extensive experiments on multiple main reasoning benchmarks show that our method achieves a significant performance improvement across different model families and sizes, thus validating its effectiveness and superiority.

## 2 RELATED WORK

**Advantage Collapse.** Advantage collapse is a critical limitation of the GRPO algorithm, as it severely impairs effective gradient updates. Prior approaches typically mitigate this issue through data filtering Yu et al. (2025); Meng et al. (2025), by discarding samples in which all responses within a group are either entirely correct or incorrect. However, this greatly limits sample efficiency, as challenging samples can be beneficial for improving model performance. In addition, some works Wang et al. (2025a) attempt to enhance response diversity by enforcing model reflection, while others Chen et al. (2025); Cheng et al. (2025) introduce internal feedback to strengthen the training signal.

**Think More or Less.** There are differing views on whether model reflection truly benefits model performance. Several works Muennighoff et al. (2025); Tian et al. (2025) proposed adding "wait" to chain-of-thought reasoning to encourage the model to engage in reflection, which can improve performance. VL-Rethinker Wang et al. (2025a) incorporates forced reflection during the training process to enhance the slow-thinking capability of the model. Meanwhile, other researchers argue that suppressing the tokens that trigger reflectionLiu et al. (2025a), encouraging the model to generate shorter responses Su et al. (2025); Fatemi et al. (2025), can reduce redundant reasoning without compromising the model's accuracy.

**RL from Internal Feedback.** Recent studies introduce internal feedback such as entropy to strengthen the training signal. Some studies Gao et al. (2025); Zhang et al. (2025a) argue that correct responses generated by models typically exhibit lower entropy than incorrect ones, so unsupervised entropy minimization methods can also enhance performance. SEED-GRPO Chen et al. (2025) introduces semantic entropy to quantify semantic diversity among generated responses and dynamically adjusts the magnitude of policy updates based on this measure. Other works Cheng et al. (2025) suggest that high entropy encourages exploratory reasoning, therefore incorporating policy entropy into the advantage term of the GRPO algorithm to promote exploration. However, most of these methods lack fine-grained modeling of the relationship between response correctness and their policy entropy.

# 3 INVESTIGATION OF ADVANTAGE COLLAPSE IN GRPO

We begin with a brief introduction to the Group Relative Policy Optimization (GRPO) algorithm Shao et al. (2024). For each input question $q$, it generates a set of responses $\{O_1, O_2, \ldots, O_G\}$ using the policy model and computes a corresponding set of rewards $\{r_1, r_2, \ldots, r_G\}$ for these responses. The rewards are then normalized to calculate the advantages. The model is optimized by maximizing the following objective function:

$$J_{GRPO}(\theta) = E_{[q,\{o_i\}]} \frac{1}{G} \sum_{i=1}^{G} \frac{1}{|o_i|} \sum_{i=1}^{|o_i|} \left\{ \min \left[ \frac{\pi_\theta}{\pi_{\theta_{old}}} A_i, \ \text{clip}\left( \frac{\pi_\theta}{\pi_{\theta_{old}}}, 1-\epsilon, 1+\epsilon \right) A_i \right] - \beta D_{KL} \right\}$$
(1)

where $\pi_\theta$ and $\pi_{\theta_{old}}$ are the current and old policy, and $A_i$ is the advantages defined as:

$$A_i = \frac{r_i - \text{mean}(\{r_1, r_2, \cdots, r_G\})}{\text{std}(\{r_1, r_2, \cdots, r_G\})}.$$
(2)

The diversity of advantages is crucial for effective model updates, as it directly determines the training signal used in policy gradient optimization. Due to the difficulty of assigning rewards to intermediate reasoning steps, most existing reward rules are sparse, with the response reward largely determined by the correctness of the final answer. As a result, when all responses within a group are either correct or incorrect, they receive identical rewards, leading to zero advantage across the group. This lack of distinction between responses impairs gradient updates, a phenomenon known as the advantage collapse problem.

Advantage collapse results in low sample efficiency. However, samples that are more challenging for the model often play an important role in improving its performance. Therefore, addressing the advantage collapse problem remains a critical challenge.

Existing approaches commonly aim to address this issue from two key perspectives: at the response level, by promoting model reflection to enhance the diversity of generated responses. At the signal level, by incorporating internal feedback mechanisms to enrich the training signal. In this work, we first conduct a preliminary investigation along both dimensions to explore their potential in mitigating the advantage collapse problem.

## 3.1 RESPONSE-LEVEL: LIMITATIONS OF REFLECTION

We begin by conducting a series of quantitative experiments to analyze the phenomenon of model reflection. We select models of two different parameter scales, including base models as well as those post-trained by supervised fine-tuning or reinforcement learning. To determine whether a

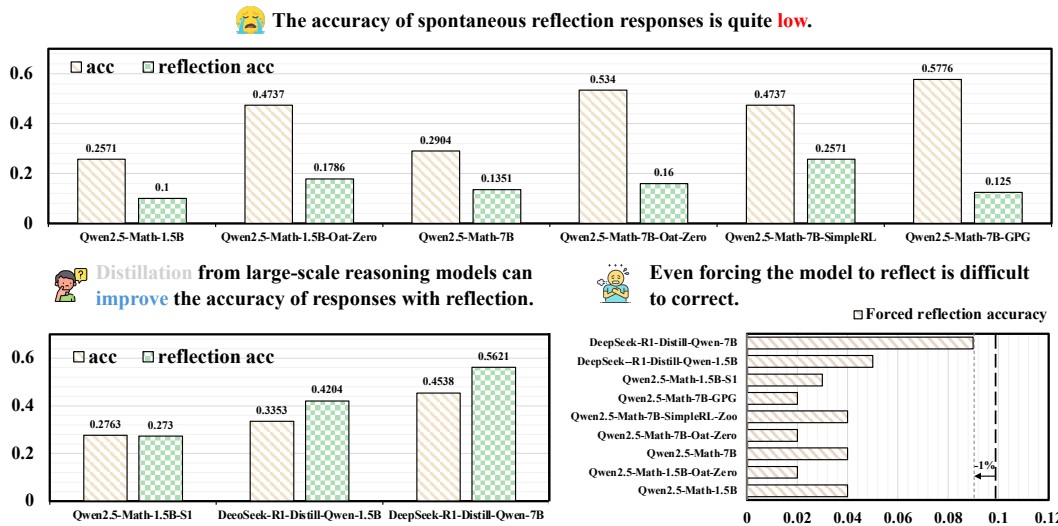

Figure 2: The reflection performance of different models. **Upper:** For most models, the accuracy of responses that involve self-reflection is significantly lower than the overall accuracy. **Left:** Fine-tuning with high-quality data that includes reflection processes helps improve the accuracy of model reflection. **Right:** Even when the model is forced to reflect on incorrect responses, the improvement in accuracy remains limited, these results are averaged over four types of reflection prompts.

model's response exhibits self-reflection, we follow previous work Liu et al. (2025b) by extracting reflection-related keywords from the responses. If the response contains reflection keywords such as `check again`, it is considered to exhibit self-reflection. The specific set of reflection keywords is provided in the Appendix.

Initially, we observe that the majority of spontaneously generated reflections by models tend to exhibit low accuracy. As shown in the upper part of Figure 2, for both base models and those post-trained by reinforcement learning, the accuracy of responses containing reflection is significantly lower than the overall accuracy of the model. This result clearly indicates that spontaneous reflection during reasoning is often ineffective and may even lead to a higher rate of incorrect responses.

However, unlike other models, two models distilled from DeepSeek-R1 Guo et al. (2025) have a more frequent self-reflection behavior, and their reflection is accompanied by higher accuracy. To verify whether this phenomenon is caused by long chain-of-thought training data from knowledge distillation, we train Qwen2.5-Math-1.5B-S1 on the S1K dataset Muennighoff et al. (2025), which contains only 1K high quality long chain-of-thought samples, some of which include reflection-related content. It can be observed that after training on the S1K data, the model's reflection accuracy significantly improves and becomes comparable to its overall accuracy.

Subsequently, we also investigate the effect of forcing different models to reflect on their incorrect answers. Specifically, we first have each model respond to every question in the test set, then we retain only the samples with incorrect answers. A reflection prompt is appended to each incorrect response to initiate reflection, after which the model is prompted to continue answering. We designed four distinct reflection prompts: `Wait!`, `Hmm`, `Let's check it again!`, and `Something is wrong here.` These prompts include two anthropomorphic expressions and two objective declarative phrases.

As shown in the lower right corner of Figure 2, the accuracy of forced reflection on incorrect responses remains below 5% for most models. Although the overall accuracy does not exceed 10%, the DeepSeek-R1-Distill series model still achieves relatively higher accuracy compared to other models due to being fine-tuned with external high-quality chain-of-thought data.

These results reveal a fundamental limitation in the reflective capabilities of most models. When model capacity is limited, relying solely on self-correction yields minimal improvement. There-fore, when confronted with challenging problems where the model persistently produces incorrect

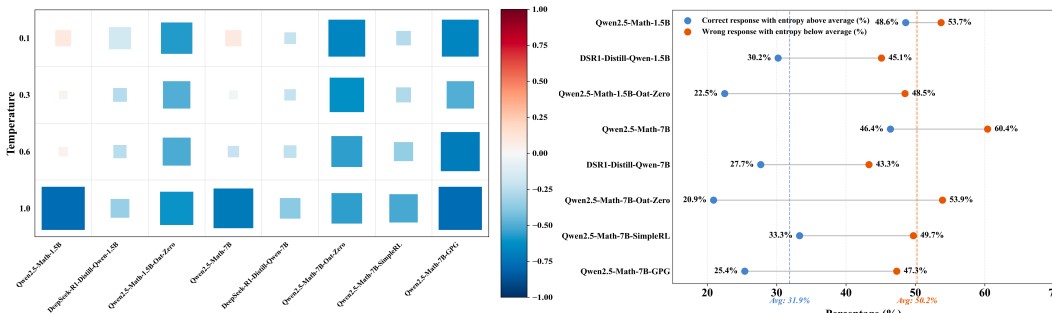

Figure 3: **Left:** The relative confidence of different models in correct responses under various temperature settings. The area of the blue squares serves as a proxy for the model's relative confidence, with larger areas reflecting greater confidence in correct responses. **Right:** The proportion of correct responses with entropy higher than the average and incorrect responses with entropy lower than the average across different models. These results are evaluated under the setting of temperature=0.1. We provide more detailed experimental results and the policy entropy distribution of different models in the Appendix.

responses, incorporating external information for correction emerges as a more effective and reliable strategy at the response level.

## 3.2 SIGNAL-LEVEL: POLICY ENTROPY

We also investigate the policy entropy of different responses. For each generated chain-of-thought response, the policy entropy $P$ is calculated as follows:

$$P = -\frac{1}{T} \sum_{t=1}^{T} \sum_{j=1}^{V} P_{t,j} \cdot \log P_{t,j}. \tag{3}$$

where $T$ denotes the total number of tokens in the response, $V$ is the vocabulary size, and $P_{t,j}$ is defined as:

$$P_{t,j} = \pi_\theta(j \in V | q, o < t) = \text{Softmax}\left(\frac{\text{logits}_t}{T}\right). \tag{4}$$

Here, $\pi_\theta$ represents the language model parameterized by $\theta$. We use the policy entropy $P$ to measure the uncertainty of the model over the generated response.

We first divide all responses into two categories based on whether they are correct or incorrect, and calculate the Relative Confidence Metric (RCM) of each model in correct responses under different temperatures using the following formula:

$$\text{RCM} = \frac{\text{Entropy}_{\text{Correct}} - \text{Entropy}_{\text{Wrong}}}{\text{Average Entropy}}. \tag{5}$$

The visualization results are shown on the left side of Figure 3. Except for the two base models, Qwen2.5-1.5B and Qwen2.5-7B Yang et al. (2024), other post-trained models typically exhibit higher relative confidence, as the average entropy of their correct responses is indeed lower than that of incorrect responses, which aligns with assumptions made in previous studies Gao et al. (2025).

However, a fine-grained analysis at the individual sample level reveals that many models exhibit miscalibrated confidence in their responses: approximately half of the incorrect responses display entropy values lower than the average, while nearly one-third of the correct responses exhibit entropy higher than the average, as shown on the right side of Figure 3. We posit that such miscalibration undermines model performance. Ideally, the model should exhibit greater confidence in the correct responses while maintaining appropriate uncertainty about its incorrect answers. Consequently, training strategies should avoid indiscriminately promoting high or low entropy, and instead adopt a more fine-grained approach that aligns policy entropy with response correctness to better guide learning dynamics.

Figure 4: The overall framework of EDGE-GRPO. By introducing Guided Error Correction at the response level to enhance response diversity and Entropy-Driven Advantage at the signal level to increase advantage diversity, we mitigate the advantage collapse problem in the vanilla GRPO. Here, $G$ represents the number of generated responses in a group.

# 4 EDGE-GRPO: ENTROPY-DRIVEN GRPO WITH GUIDED ERROR CORRECTION

Building on the above insights, we propose the EDGE-GRPO algorithm by introducing Guided Error Correction (GEC) to enhance response diversity and Entropy-Driven Advantage (EDA) to augment signal diversity, thus addressing the advantage collapse problem.

## 4.1 RESPONSE-LEVEL: GUIDED ERROR CORRECTION (GEC)

The experimental analysis in the previous section has shown that the model's ability to correct errors through reflection is quite limited. This limitation leads the model to consistently generate entirely incorrect responses when faced with problems beyond its capabilities. However, response diversity fundamentally impacts reward diversity, which consequently directly affects advantage diversity. Therefore, introducing external solutions to ensure that each set of responses contains a certain proportion of positive and negative samples is crucial for mitigating the advantage collapse problem.

To fundamentally address this issue, we propose Guided Error Correction (GEC), a response-level intervention strategy designed to mitigate advantage collapse by enhancing response diversity. As illustrated in Figure 4, for incorrect responses, GEC performs one of the following three operations based on a predefined probability:

**Reference Solution Replacement:** The incorrect response is completely replaced with an external reference solution. This operation is performed with a probability of $P_1 = \frac{2}{G}$, where $G$ is the total number of responses in the group. Compared to a smaller value ($P_1 = \frac{1}{G}$), this value ensures a higher probability that at least one response will be replaced with the reference solution when all responses are incorrect. Compared to a higher probability, it also maintains response diversity and avoids having too many identical responses within the group.

**Direct Answer Injection:** Along with the reflection prompt, the correct answer is provided directly. This operation is also performed with a probability of $P_2 = \frac{2}{G}$ to ensure both the number and the diversity of correct responses within the group.

**Prompt and Regenerate:** A simple reflection prompt is provided, and the model is asked to regenerate its answer based on it, giving the model a chance to self-correct. This operation is performed

Table 1: Pass@1 performance comparison across various mathematical evaluation benchmarks. The results below are from 1 epoch of training on **DeepScaleR-Hard-1K**. The number of samples in each benchmark is indicated in parentheses. The results are evaluated under the setting of temperature = 0.1. The best results are indicated by **boldface**.

| Model | Method | Avg (1560) | AIME (30) | AMC (83) | Math (500) | Min (272) | Oly (675) |
|---|---|---|---|---|---|---|---|
| Qwen2.5-Math-1.5B | Base | 25.71 | 6.67 | 37.35 | 34.60 | 12.13 | 24.00 |
| | SFT | 29.17 | 6.67 | 28.92 | 46.40 | 12.87 | 24.00 |
| | Vanilla GRPO | 40.26 | 10.00 | **46.99** | 65.00 | 20.59 | 30.37 |
| | + Force Reflection | 41.55 | **13.33** | 31.33 | 70.00 | 22.06 | 30.81 |
| | Dr.GRPO | 40.13 | 13.33 | 43.37 | 67.20 | 18.75 | 29.48 |
| | DAPO | 33.91 | 6.67 | 31.33 | 54.80 | 16.54 | 26.96 |
| | **EDGE-GRPO** | **47.24** | 10.00 | 44.58 | **73.20** | **29.04** | **37.33** |
| Qwen2.5-Math-7B | Base | 29.04 | 10.00 | 37.35 | 53.40 | 10.66 | 18.22 |
| | SFT | 37.37 | 3.33 | 44.58 | 68.00 | 19.85 | 22.37 |
| | Vanilla GRPO | 47.69 | **26.67** | **53.01** | 74.20 | 25.74 | 37.19 |
| | + Force Reflection | 40.26 | 13.33 | 40.96 | 67.60 | 19.85 | 29.33 |
| | Dr.GRPO | 48.78 | 23.33 | 56.63 | 75.20 | 27.21 | 38.07 |
| | DAPO | 49.49 | 20.00 | 57.83 | 76.20 | 27.94 | 38.67 |
| | **EDGE-GRPO** | **53.21** | 16.67 | **53.01** | **79.00** | **36.03** | **42.67** |
| Llama-3.2-3B-Instruct | Base | 19.81 | 6.67 | 14.46 | 36.20 | 12.13 | 12.00 |
| | SFT | 22.44 | 0.00 | 20.48 | 41.20 | 11.77 | 14.07 |
| | Vanilla GRPO | 22.44 | **13.33** | 15.66 | 42.80 | 15.07 | 11.56 |
| | + Force Reflection | 22.89 | 6.67 | **21.69** | 41.8 | 13.97 | 13.33 |
| | Dr.GRPO | 22.24 | 3.33 | 18.07 | 42.60 | 12.50 | 12.44 |
| | DAPO | 22.95 | 3.33 | 18.07 | 42.60 | 13.97 | 13.48 |
| | **EDGE-GRPO** | **25.06** | 3.33 | 20.48 | **45.60** | **17.28** | **14.52** |

with a probability of $P_3 = 1 - P_1 - P_2$, which ensures that most responses are still generated by the model itself. Since the effect of self-reflection is relatively limited, only a small portion of the responses can be corrected to the right answers. These few corrected responses are then used as positive samples to guide the model update.

These three strategies ensure that each group of responses contains positive samples with correct answers while still retaining negative samples generated by the model itself.

By introducing Guided Error Correction at the response level, we ensure that even when the model encounters problems beyond its capabilities, the response set can still contain diverse answers. This helps mitigate the issue of advantage collapse and provides effective training signals.

## 4.2 SIGNAL-LEVEL: ENTROPY-DRIVEN ADVANTAGE (EDA)

Although Guided Error Correction enhances response diversity and prevents the advantages within a group from collapsing to zero, it remains insufficient to address the issue of uniform advantages among correct or incorrect responses. To enable finer-grained differentiation among different correct or incorrect responses, we introduce policy entropy as an internal feedback signal to enhance advantage diversity.

The results in the previous section show that the model often misjudges the confidence of its responses, many incorrect responses exhibit low entropy, while many correct responses have high entropy. We believe this misalignment negatively impacts model performance. Therefore, we propose Entropy-Driven Advantage (EDA) to enhance the model's ability to distinguish between different responses.

For each response $O_i$ during training, we calculate its policy entropy $P_i$ using Equations 3 and 4, and then scale it to ensure the values remain within a reasonable range.

$$\hat{P}_i = \frac{P_i}{\text{mean}(\{P_1, P_2, \cdots, P_G\})}.$$ (6)

Next, we use the scaled entropy values to compute the entropy-driven advantage:

$$\hat{A}_i = \frac{r_i - \text{mean}(\{r_1, r_2, \cdots, r_G\})}{\text{std}(\{r_1, r_2, \cdots, r_G\}) \cdot \hat{P}_i}.$$ (7)

Compared to the initial advantage values, the entropy-driven advantage exhibits greater diversity. It assigns higher advantages to responses that are both correct and confident, while imposing harsher

penalties on responses that are incorrect but overly confident. It ensures that, when the initial advantages are not all zero, different responses are assigned distinct final advantage values, thereby enhancing the model's ability to distinguish among responses and further mitigating the advantage collapse problem.

It is worth emphasizing that the GEC and EDA modules enhance the diversity of advantage from different levels, and they are complementary to some extent. Although the GEC module ensures that the advantage of the response is not zero, it cannot achieve finer-grained differentiation between correct or incorrect responses. In contrast, the EDA module relies on the presence of a certain number of positive and negative samples within the group. When the initial intra-group advantages are all zero, fine-grained scaling cannot take effect. This is precisely the reason for incorporating a proportion of correct responses into the GEC module.

## 5 EXPERIMENTS

### 5.1 EXPERIMENTAL SETUP

**Train Datasets.** We use the DeepScaleR dataset Luo et al. (2025) for training. The original dataset contains approximately 40K math problems. We retain only those samples that include a solution and where the final answer is placed inside a \boxed{} in the solution. After this filtering process, around 2K samples remain. We randomly select 1K samples as the standard training set, named DeepScaleR-Random-1K. Meanwhile, to evaluate the effectiveness of our method on more challenging data, we use Qwen2.5-Math-7B to further filter the samples. Specifically, for each question, the model generates eight responses, and we select the 1K questions with the lowest accuracy as the hard training set, referred to as DeepScaleR-Hard-1K. In this dataset, approximately 80% of the questions receive entirely incorrect responses across all generations.

**Evaluation Benchmark.** We select five challenging mathematical reasoning benchmarks to evaluate our method: AIME24, AMC, MATH500 Hendrycks et al. (2021), Minerva Lewkowycz et al. (2022) and OlympiadBench He et al. (2024). These benchmarks collectively contain a total of 1,560 problems. All evaluation experiments in this paper are conducted on these benchmarks.

**Implementation Details.** We conduct experiments on 8 NVIDIA A100-40G GPUs. We remove the KL divergence to eliminate constraints on the model. Previous studies Yu et al. (2025); Liu et al. (2025b) have shown that it can lead to better training performance, as the distribution of the model may differ significantly from the initial model during training. Other training configurations and hyperparameter settings follow the default setup of the GRPO trainer under the TRL framework von Werra et al. (2020). We train for one epoch on only 1K DeepScaleR samples on Qwen2.5-Math-1.5B, Qwen2.5-Math-7B Yang et al. (2024) and Llama3.2-3B-Instruct Grattafiori et al. (2024).

During evaluation, we focus on the model's pass@1 performance, meaning the model generates only one response for each given question. To calculate the overall average accuracy, we avoid directly averaging the accuracy across the five benchmarks due to their varying number of questions. Instead, we calculate the average by dividing the total number of correct answers by the total number of questions to reduce bias. More detailed experimental settings can be found in the Appendix.

### 5.2 MAIN RESULT

Table 1 presents the results of our method on various mathematical evaluation benchmarks. Although our method is trained on only 1K samples for one epoch, it achieves significant performance improvements across various scales of Qwen and Llama models. It is worth emphasizing that for about 80% of the problems in DeepScaleR-Hard-1K, all eight responses generated by the base model Qwen2.5-Math-7B are incorrect. This further validates the effectiveness of our method in challenging data, even when the difficulty exceeds the capability of the model. In addition, we also conduct experiments on DeepScaleR-Random-1K to verify the generalization performance of our method, with detailed results provided in the Appendix.

Since our method requires each question to have not only the final answer, but also a corresponding reference solution, we established a Supervised Fine-Tuning (SFT) baseline using the same chain-

Table 2: The ablation study of EDGE-GRPO separately verifies the effectiveness of guided error correction and entropy-driven advantage. The results are all from training on DeepScaleR-Hard-1K.

| Method | Avg (1560) | AIME (30) | AMC (83) | Math (500) | Min (272) | Oly (675) |
|---|---|---|---|---|---|---|
| **EDGE-GRPO [Qwen2.5-Math-1.5B]** | 47.24 | 10.00 | 44.58 | 73.20 | 29.04 | 37.33 |
| - Reference Solution | 42.44 | 10.00 | 34.94 | 69.60 | 23.53 | 32.30 |
| - Guided Error Correction | 40.06 | 13.33 | 40.96 | 64.00 | 21.32 | 30.96 |
| - Entropy-Driven Advantage | 40.64 | 10.00 | 43.37 | 67.80 | 19.12 | 30.22 |
| **EDGE-GRPO [Qwen2.5-Math-7B]** | 53.21 | 16.67 | 53.01 | 79.00 | 36.03 | 42.67 |
| - Reference Solution | 45.19 | 26.67 | 50.60 | 68.00 | 28.68 | 35.11 |
| - Guided Error Correction | 46.80 | 20.00 | 50.60 | 75.80 | 23.16 | 35.56 |
| - Entropy-Driven Advantage | 47.44 | 3.33 | 43.37 | 74.40 | 29.04 | 37.33 |

of-thought data for comparison. As shown in the SFT row of Table 1, our approach significantly outperforms supervised fine-tuning, even when trained on the exact same data. And our method also exhibits a clear advantage when evaluated against established algorithmic variants. Specifically, our method consistently outperforms algorithmic variants such as Dr.GRPO Liu et al. (2025b) and DAPO Yu et al. (2025), as well as improvements to the vanilla GRPO algorithm that force the model to reflect on incorrect responses.

In addition, we conduct an ablation study on EDGE-GRPO, as shown in Table 2. First, removing external reference solutions leads to a decline in model performance, highlighting the importance of integrating external information when dealing with problems beyond the model's capability. In addition, removing either the GEC or EDA component results in a significant performance gap compared to EDGE-GRPO, which not only validates the effectiveness of each component but also underscores the importance of enhancing response diversity and providing more fine-grained distinctions in training signals for improving model performance.

We also visualize changes in advantage variance during training for different methods, as shown in Figure 5. During training, compared with the vanilla GRPO algorithm and the variant with enforced reflection, our method maintains a higher level of intra-group advantage variance solely through response-level improvements (GEC). After advantage scaling, advantage diversity will be further enhanced. It also demonstrates its significant mitigation of the advantage collapse problem.

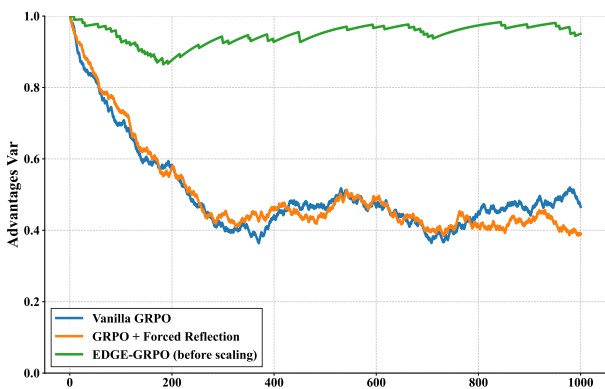

Figure 5: The changes in intra-group advantage variance during training for different methods. Our method maintains a relatively high level without significant decline.

Moreover, even with significantly less training data, our model achieves comparable performance to other main open source models Zeng et al. (2025); Hu et al. (2025); Cui et al. (2025); Liu et al. (2025b), as illustrated in Figure 1. This further demonstrates the superiority and utility of our method. More detailed experimental results can be found in the Appendix.

## 6 CONCLUSION

This work proposes a simple and effective EDGE-GRPO algorithm that mitigates the advantage collapse problem of the vanilla GRPO algorithm on two levels. At the response level, the Guided Error Correction (GEC) method is introduced to overcome the limitations of the inherent capabilities of the model and improve response diversity. At the signal level, the Entropy-Driven Advantage (EDA) computation enables the model to differentiate responses more finely during training, thereby improving the diversity of advantages. Our method significantly alleviates the advantage collapse problem and achieves notable performance improvements using only 1K samples across different base models, demonstrating its effectiveness and superiority.

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

# A    APPENDIX

## A.1    REFLECTION KEYWORDS

We determine whether a reflection phenomenon has occurred based on the presence of reflection-related keywords in the response. The set of 15 keywords used in the experiment for identifying reflection is as follows: `check again`, `recheck`, `double-check`, `rethink`, `think again`, `reevaluate`, `re-evaluate`, `re-examine`, `verify again`, `reevaluation`, `reexamine`, `reanalyze`, `reassess`, `reconsider`, `go over`.

## A.2    ANALYSIS ON THE INTRODUCTION OF EXTERNAL SOLUTIONS

In the Guided Error Correction (GEC) section, we introduce external reference solutions to enhance response diversity. Since the reference solution trajectories are not sampled by the old policy, our method is not strictly on-policy. So we approximate the probability of these solutions under the old model policy $\pi_{\theta_{old}}$ using the probability that the old model would generate the trajectories of these reference solutions. The specific probability calculation is shown in Algorithm 1.

In practice, we consider filtering out reference solutions whose trajectory probability under $\pi_{\theta_{old}}$ was below a certain threshold to reduce variance and stabilize model training. However, we observed that during training with external reference solutions, the model gradient updates remain stable, as shown in Figure 6, so we ultimately decided to keep all the reference solutions.

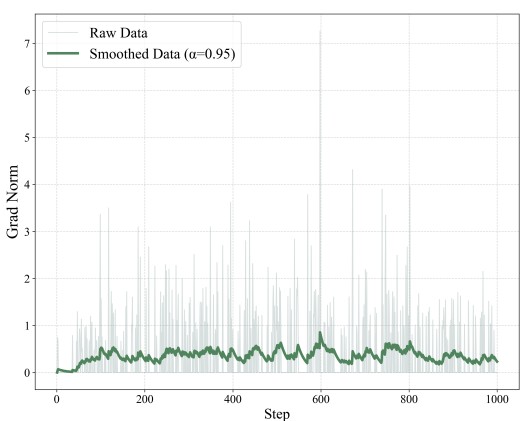

Figure 6: The changes in gradient during training after introducing reference solution.

---

**Algorithm 1** GRPO with Reference Solution Replacement

---

1: **Input:** Prompt $p$, Reference answer $a^*$, Reference solution $s^*$, Number of generations $G$
2: **// 1. Generate and Check**
3: **for** $i = 1$ to $G$ **do**
4:     $c_i \leftarrow$ model.generate$(p)$, $r_i \leftarrow$ reward$(p, c_i)$
5:     **if** $r_i \neq 2$ **then**                                    ▷ Incorrect response
6:         With $\frac{2}{G}$ probability: $c_i \leftarrow s^*$          ▷ Replace with reference solution
7:     **end if**
8: **end for**
9: **// 2. Compute Logits**
10: **for** $i = 1$ to $G$ **do**
11:     tokens$_i \leftarrow$ tokenize$(p + c_i)$
12:     $\log p_i^\pi \leftarrow$ model_logprobs(tokens$_i, \tau$)                ▷ Current policy
13:     $\log p_{old,i}^\pi \leftarrow \log p_i^\pi$.detach()                  ▷ Old policy
14: **end for**
15: **// 3. GRPO Loss**
16: **for** $i = 1$ to $G$ **do**
17:     $A_i \leftarrow$ Normalize$(r_i - \bar{r})$, $\rho_i \leftarrow \exp(\log p_i^\pi - \log p_{old,i}^\pi)$
18:     $\mathcal{L}_i \leftarrow -\min(\rho_i, \text{clip}(\rho_i, 1 - \epsilon, 1 + \epsilon)) \cdot A_i$
19: **end for**
20: $\mathcal{L} \leftarrow \frac{1}{G} \sum_{i=1}^{G} \mathcal{L}_i$
21: **return** $\mathcal{L}$

---

## A.3 Detailed Experimental Settings

Our training configuration and hyperparameter settings follow the default settings of the GRPO trainer under the TRL framework. For each sample, the model is prompted to generate 8 responses, each response limited to a maximum of 1024 tokens. We train on 1K samples per epoch, and to enable experiments on Qwen2.5-Math-7B, we generate one response per GPU, resulting in a total of 1K training steps. The learning rate is set to 1e-6 during training. For evaluation on the five reasoning benchmarks, all tests are conducted with a temperature setting of 0.1.

## A.4 Detailed Experimental Results

### A.4.1 Further experimental results

We also conduct experiments on DeepScaleR-Random-1K, with results shown in Table 3. Our method achieves performance improvements across all three different base models. Furthermore, compared to algorithm variants such as Dr.GRPO and DAPO, as well as methods that enforce reflection on the vanilla GRPO, our method demonstrates clear advantages. We also observe that the performance improvement of Qwen2.5-Math-7B is more limited compared to training on DeepScaleR-Hard-1K. We attribute this to the fact that this training data is relatively easier for the model, which further highlights the effectiveness of our method on difficult data, as such challenging data often contributes more to performance improvement.

Table 3: Pass@1 performance comparison across various mathematical evaluation benchmarks. The results below are from 1 epoch of training on **DeepScaleR-Random-1K**. The number of samples in each benchmark is indicated in parentheses. The results are evaluated under the setting of temperature = 0.1. The best results are indicated by **boldface**.

| Model | Method | Avg (1560) | AIME (30) | AMC (83) | Math (500) | Min (272) | Oly (675) |
|-------|--------|------------|-----------|----------|------------|-----------|-----------|
| Qwen2.5-Math-1.5B | Base | 25.71 | 6.67 | 37.35 | 34.60 | 12.13 | 24.00 |
| | SFT | 30.13 | 10.00 | 30.12 | 47.20 | 14.71 | 24.59 |
| | Vanilla GRPO | 40.32 | **13.33** | 39.76 | 65.60 | 19.49 | 31.26 |
| | + Force-R | 42.63 | 10.00 | 36.14 | 71.40 | 21.69 | 32.00 |
| | Dr.GRPO | 40.39 | 6.67 | 40.96 | 66.80 | 20.22 | 30.37 |
| | DAPO | 41.67 | 10.00 | **44.58** | 67.80 | 19.49 | 32.30 |
| | EDGE-GRPO | **48.08** | 13.33 | **44.58** | **76.40** | **28.68** | **36.89** |
| Qwen2.5-Math-7B | Base | 29.04 | 10.00 | 37.35 | 53.40 | 10.66 | 18.22 |
| | SFT | 41.99 | 6.67 | 43.37 | 69.00 | 22.06 | 31.41 |
| | Vanilla GRPO | 46.47 | **23.33** | 55.42 | 72.40 | 27.94 | 34.67 |
| | + Force-R | 47.76 | **23.33** | 53.01 | 74.60 | 23.16 | **38.22** |
| | Dr.GRPO | 48.78 | 16.67 | 53.01 | **75.60** | 29.04 | 37.78 |
| | DAPO | 46.99 | 20.00 | **59.04** | 73.00 | 26.84 | 35.56 |
| | EDGE-GRPO | **49.30** | 16.67 | 50.60 | **75.60** | **33.09** | 37.04 |
| Llam3.2-3B-Instruct | Base | 19.81 | 6.67 | 14.46 | 36.20 | 12.13 | 12.00 |
| | SFT | 23.46 | 0.00 | 18.07 | 43.40 | 14.71 | 13.93 |
| | Vanilla GRPO | 24.49 | **13.33** | 18.07 | 45.40 | 15.44 | 13.93 |
| | + Force-R | 23.72 | 10.00 | 16.87 | 44.00 | 12.87 | 14.52 |
| | Dr.GRPO | 22.89 | **13.33** | **24.10** | 43.20 | 12.50 | 12.30 |
| | DAPO | 21.22 | 6.67 | 9.64 | 40.80 | 13.24 | 12.00 |
| | EDGE-GRPO | **25.90** | 3.33 | 19.28 | **47.60** | **17.28** | **15.11** |

### A.4.2 Performance Comparison During Training

To visually demonstrate the superiority of our method during the training process, Figure 7 plots the performance curves of our method, the method with only forced reflection added to the vanilla GRPO (GRPO + Forced reflection), and the original GRPO method on multiple benchmark during training steps.

As can be clearly seen from the figure, our method consistently and significantly outperforms the other two baseline throughout the entire training process. In comparison, the GRPO method with only forced reflection shows some improvement over the vanilla GRPO, but the effect is limited.

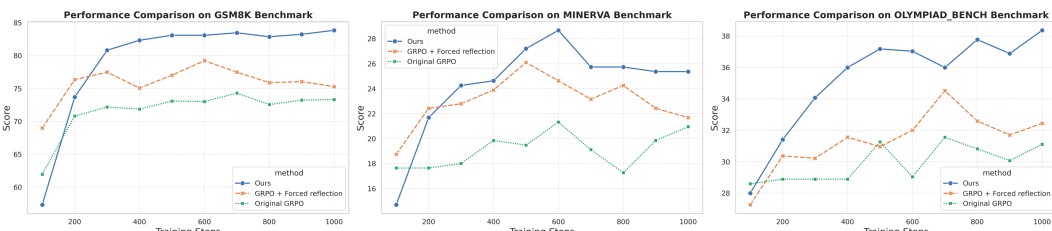

Figure 7: Performance comparison of different methods on three benchmarks during training steps. Our method consistently outperforms the vanilla GRPO and the variant with forced reflection throughout the training process.

### A.4.3 COMPARE WITH OTHER OPEN SOURCE MODELS

To more comprehensively evaluate the effectiveness of our method, we conducted a comparison of our models against current mainstream open source models on Pass@1 performance across five mathematical reasoning benchmarks. Detailed comparison results are presented in Table 4. A core highlight is that our method achieves excellent results with extremely high data efficiency. Our models were trained using only 1K selected samples, far smaller than some other models which require tens of thousands of samples or more.

Table 4: Pass@1 performance comparison of our models against various open-source models on five mathematical reasoning benchmarks. Our models, trained on only 1K samples, demonstrate highly competitive performance. The total number of problems for each benchmark is indicated in parentheses. * denotes data from the original paper, other results are from our own evaluation.

| Model | # Train | Avg (1560) | AIME (30) | AMC (83) | Math (500) | Min (272) | Olym (675) |
|---|---|---|---|---|---|---|---|
| Qwen2.5-Math-1.5B | Base | 25.71 | 6.67 | 37.35 | 34.60 | 12.13 | 24.00 |
| DeepSeek-Distill-1.5B | 800K | 33.53 | 6.67 | 27.71 | 61.00 | 13.60 | 23.11 |
| Oat-Zero-1.5B | 8.5K | 47.37 | 20.00 | 48.19 | 75.00 | 25.74 | 36.74 |
| Qwen2.5-Math-7B | Base | 29.04 | 10.00 | 37.35 | 53.40 | 10.66 | 18.22 |
| DeepSeek-Distill-7B | 800K | 45.39 | 16.67 | 36.14 | 74.20 | 29.41 | 32.89 |
| Oat-Zero-7B | 8.5K | 53.40 | 36.67 | 61.45 | 79.80 | 30.88 | 42.67 |
| SimpleRL-Zoo-7B | 8K | 47.37 | 23.33 | 53.01 | 76.00 | 24.26 | 35.85 |
| Eurus-7B[*] | 48.4K | 53.9 | 26.7 | 57.8 | 79.2 | 38.6 | 42.1 |
| OpenReasoner-Zero-7B | 5.7K | 51.99 | 20.00 | 40.96 | 80.20 | 29.41 | 42.96 |
| **EDGE-GRPO-1.5B** | 1K | 48.08 | 13.33 | 44.58 | 76.40 | 28.68 | 36.89 |
| **EDGE-GRPO-7B** | 1K | 53.01 | 16.67 | 49.40 | 79.00 | 36.03 | 42.67 |

### A.5 DETAILED RESULTS IN REFLECTION

This section provides a more detailed quantitative analysis of the model reflection phenomenon, which we examined from the perspectives of both spontaneous and forced reflection.

Table 5 analyzes the spontaneous reflection behavior of the models. The results show that for most models, the accuracy of responses involving spontaneous reflection is significantly lower than their overall average accuracy and the accuracy of responses without reflection. A notable exception, however, is the DeepSeek-R1-Distill series of models, which were distilled from large reasoning models. Their reflection accuracy is much higher than their average on the contrary, corroborating the point made in the main text that high-quality knowledge distillation helps improve effective reflection capabilities. Furthermore, the Qwen2.5-Math-1.5B-S1 model, trained on high-quality chain-of-thought data distilled from large reasoning models, also exhibits a reflection accuracy close to its overall accuracy, outperforming most other models.

Table 6 investigates the effect of forced reflection on the other hand. We selected samples where the models provided incorrect answers and forced them to reflect and correct their responses using four different prompts. The results show that for the vast majority of models, the improvement in accuracy from forced reflection is very limited, with correction accuracy rates generally below 10%. Even for the top-performing Deepseek-R1-Distill-Qwen-7B, the highest correction rate is only around 11%.

Table 5: Model performance analysis. The table compares overall accuracy, accuracy on samples with reflection and accuracy on samples without reflection.

| Temperature | Model | Average Acc | Reflection Acc | No-Reflection Acc |
|---|---|---|---|---|
| 0.1 | Qwen2.5-Math-1.5B | 25.71 | 10.00 | 26.01 |
| | DeepSeek-R1-Distill-Qwen-1.5B | 33.53 | 42.04 | 14.37 |
| | Qwen2.5-Math-1.5B-S1 | 27.63 | 27.30 | 27.93 |
| | Qwen2.5-Math-1.5B-Oat-Zero | 47.37 | 17.86 | 47.91 |
| | Qwen2.5-Math-7B | 29.04 | 13.51 | 29.42 |
| | DeepSeek-R1-Distill-Qwen-7B | 45.38 | 56.21 | 22.77 |
| | Qwen2.5-Math-7B-Oat-Zero | 53.40 | 16.00 | 54.01 |
| | Qwen2.5-Math-7B-SimpleRL | 47.37 | 25.71 | 47.87 |
| | Qwen2.5-Math-7B-GPG | 57.76 | 12.50 | 58.23 |
| 0.3 | Qwen2.5-Math-1.5B | 25.51 | 20.00 | 25.66 |
| | DeepSeek-R1-Distill-Qwen-1.5B | 33.27 | 41.47 | 16.44 |
| | Qwen2.5-Math-1.5B-S1 | 26.86 | 26.73 | 26.97 |
| | Qwen2.5-Math-1.5B-Oat-Zero | 48.65 | 15.15 | 49.38 |
| | Qwen2.5-Math-7B | 26.28 | 14.81 | 26.48 |
| | DeepSeek-R1-Distill-Qwen-7B | 45.77 | 56.32 | 25.38 |
| | Qwen2.5-Math-7B-Oat-Zero | 52.88 | 16.22 | 53.78 |
| | Qwen2.5-Math-7B-SimpleRL | 48.14 | 22.86 | 48.72 |
| | Qwen2.5-Math-7B-GPG | 57.82 | 10.53 | 58.40 |
| 0.6 | Qwen2.5-Math-1.5B | 20.00 | 7.55 | 20.44 |
| | DeepSeek-R1-Distill-Qwen-1.5B | 34.04 | 41.71 | 22.19 |
| | Qwen2.5-Math-1.5B-S1 | 20.32 | 24.82 | 16.65 |
| | Qwen2.5-Math-1.5B-Oat-Zero | 46.92 | 13.79 | 47.55 |
| | Qwen2.5-Math-7B | 20.77 | 9.09 | 21.02 |
| | DeepSeek-R1-Distill-Qwen-7B | 44.23 | 54.27 | 25.99 |
| | Qwen2.5-Math-7B-Oat-Zero | 53.08 | 14.29 | 53.97 |
| | Qwen2.5-Math-7B-SimpleRL | 45.45 | 17.50 | 46.18 |
| | Qwen2.5-Math-7B-GPG | 54.23 | 8.33 | 54.95 |
| 1 | Qwen2.5-Math-1.5B | 11.35 | 8.57 | 11.48 |
| | DeepSeek-R1-Distill-Qwen-1.5B | 27.44 | 40.06 | 16.65 |
| | Qwen2.5-Math-1.5B-S1 | 20.83 | 21.95 | 19.93 |
| | Qwen2.5-Math-1.5B-Oat-Zero | 45.51 | 4.17 | 46.16 |
| | Qwen2.5-Math-7B | 16.92 | 5.80 | 17.44 |
| | DeepSeek-R1-Distill-Qwen-7B | 41.41 | 51.71 | 30.87 |
| | Qwen2.5-Math-7B-Oat-Zero | 52.50 | 21.62 | 53.25 |
| | Qwen2.5-Math-7B-SimpleRL | 45.38 | 20.93 | 46.08 |
| | Qwen2.5-Math-7B-GPG | 49.81 | 6.06 | 50.75 |

Collectively, this data indicates that a significant bottleneck persists in the self-correction capabilities of existing models through reflection, whether spontaneous or forced. This supports the necessity of our proposed Guided Error Correction (GEC) method.

## A.6 DETAILED RESULTS IN POLICY ENTROPY

This section quantitatively analyzes the relationship between model confidence and answer correctness using the Relative Confidence Metric (RCM). The RCM is calculated as:

$$\text{RCM} = \frac{\text{Entropy}_{\text{Correct}} - \text{Entropy}_{\text{Incorrect}}}{\text{Average Entropy}}$$

As shown in Table 7, a negative RCM value indicates that the model's correct responses have a lower average entropy than its incorrect ones, meaning the model is more confident in its correct answers than incorrect ones.

The results show that models can better calibrate their confidence on the whole, expressing higher confidence in correct answers than in incorrect ones. However, the RCM is an aggregate metric that reflects a macroscopic trend. The model still exhibits a significant number of high-confidence incorrect responses at the individual sample level. Therefore, we propose our Entropy-Driven Advantage (EDA) to apply more fine-grained rewards and penalties at the signal level.

To visually substantiate our analysis of policy entropy at a fine-grained level, Figure 8 presents the entropy distributions of correct and incorrect responses across a variety of models, including base models and those enhanced through different post-training methods.

Table 6: Reflection accuracy under different reflection triggers. The "Incorrect" column shows the total number of wrong answers. The subsequent columns show the reflection accuracy scores for specific trigger words.

| Temperature | Model | Incorrect | Reflection accuracy (%) | | | |
|---|---|---|---|---|---|---|
| | | | Wait! | Hmm | Let's check it | Something is wrong |
| 0.1 | Qwen2.5-Math-1.5B | 1159 | 3.624 | 2.675 | 7.161 | 3.365 |
| | DeepSeek-R1-Distill-Qwen-1.5B | 1037 | 4.638 | 5.700 | 5.507 | 4.251 |
| | Qwen2.5-Math-1.5B-Oat-Zero | 821 | 2.314 | 1.462 | 0.974 | 2.923 |
| | Qwen2.5-Math-7B | 1107 | 5.872 | 5.059 | 1.987 | 4.426 |
| | DeepSeek-R1-Distill-Qwen-7B | 852 | 8.706 | 11.059 | 8.588 | 8.353 |
| | Qwen2.5-Math-7B-Oat-Zero | 727 | 3.026 | 0.688 | 0.413 | 1.926 |
| | Qwen2.5-Math-7B-SimpleRL-Zoo | 821 | 5.366 | 4.146 | 1.098 | 5.122 |
| | Qwen2.5-Math-7B-GPG | 659 | 4.401 | 1.517 | 0.152 | 2.731 |
| 0.3 | Qwen2.5-Math-1.5B | 1162 | 3.184 | 3.356 | 5.594 | 4.389 |
| | DeepSeek-R1-Distill-Qwen-1.5B | 1041 | 5.967 | 7.507 | 8.277 | 5.101 |
| | Qwen2.5-Math-1.5B-Oat-Zero | 801 | 1.748 | 1.623 | 0.250 | 1.873 |
| | Qwen2.5-Math-7B | 1150 | 5.826 | 4.435 | 3.217 | 2.696 |
| | DeepSeek-R1-Distill-Qwen-7B | 846 | 10.308 | 9.597 | 8.649 | 9.123 |
| | Qwen2.5-Math-7B-Oat-Zero | 735 | 3.129 | 1.769 | 0.136 | 3.401 |
| | Qwen2.5-Math-7B-SimpleRL-Zoo | 809 | 6.057 | 3.585 | 0.865 | 6.057 |
| | Qwen2.5-Math-7B-GPG | 658 | 2.888 | 1.216 | 0.152 | 3.495 |
| 0.6 | Qwen2.5-Math-1.5B | 1248 | 2.648 | 2.809 | 5.056 | 3.852 |
| | DeepSeek-R1-Distill-Qwen-1.5B | 1029 | 7.101 | 7.879 | 6.323 | 6.323 |
| | Qwen2.5-Math-1.5B-Oat-Zero | 828 | 2.053 | 1.208 | 0.725 | 1.932 |
| | Qwen2.5-Math-7B | 1236 | 4.288 | 3.722 | 1.861 | 5.502 |
| | DeepSeek-R1-Distill-Qwen-7B | 870 | 11.406 | 12.097 | 11.290 | 10.253 |
| | Qwen2.5-Math-7B-Oat-Zero | 732 | 2.869 | 1.093 | 0.137 | 3.005 |
| | Qwen2.5-Math-7B-SimpleRL-Zoo | 851 | 4.935 | 4.113 | 3.055 | 4.465 |
| | Qwen2.5-Math-7B-GPG | 714 | 3.922 | 1.821 | 0.280 | 2.801 |
| 1 | Qwen2.5-Math-1.5B | 1383 | 1.952 | 1.735 | 2.531 | 1.591 |
| | DeepSeek-R1-Distill-Qwen-1.5B | 1132 | 9.637 | 10.610 | 9.637 | 8.753 |
| | Qwen2.5-Math-1.5B-Oat-Zero | 850 | 1.765 | 1.882 | 0.471 | 1.882 |
| | Qwen2.5-Math-7B | 1296 | 1.931 | 2.008 | 1.236 | 1.776 |
| | DeepSeek-R1-Distill-Qwen-7B | 914 | 9.430 | 9.649 | 9.320 | 9.539 |
| | Qwen2.5-Math-7B-Oat-Zero | 741 | 2.699 | 1.350 | 0.405 | 2.699 |
| | Qwen2.5-Math-7B-SimpleRL-Zoo | 852 | 3.169 | 2.582 | 0.235 | 2.347 |
| | Qwen2.5-Math-7B-GPG | 783 | 1.788 | 1.660 | 0.255 | 1.788 |

Each plot within the figure displays two overlapping density distributions: one for correct responses and another for incorrect responses. These graphs visually confirm that most models exhibit misjudgments in their response confidence. A consistent pattern is the substantial overlap between the entropy distributions of correct and incorrect answers.

Specifically, these visualizations reveal two key phenomena: Firstly, a considerable portion of incorrect responses possesses low entropy, indicating that the models are often highly confident in

Table 7: Relative Confidence Metric (RCM) across different models and temperature settings. A negative value indicates that the model exhibits lower entropy (i.e., higher confidence) in its correct responses compared to its incorrect responses.

| Model | Temperature | | | |
|---|---|---|---|---|
| | 0.1 | 0.3 | 0.6 | 1.0 |
| Qwen2.5-Math-1.5B | 0.0909 | 0.0278 | 0.0349 | -0.7773 |
| DeepSeek-R1-Distill-Qwen-1.5B | -0.1667 | -0.2596 | -0.2521 | -0.3476 |
| Qwen2.5-Math-1.5B-Oat-Zero | -0.5714 | -0.5000 | -0.5116 | -0.6049 |
| Qwen2.5-Math-7B | 0.0909 | -0.0294 | -0.2171 | -0.7175 |
| DeepSeek-R1-Distill-Qwen-7B | -0.2222 | -0.2195 | -0.2406 | -0.3808 |
| Qwen2.5-Math-7B-Oat-Zero | -0.6667 | -0.6250 | -0.5625 | -0.5556 |
| Qwen2.5-Math-7B-SimpleRL | -0.2727 | -0.2813 | -0.3553 | -0.5179 |
| Qwen2.5-Math-7B-GPG | -0.6667 | -0.5000 | -0.7059 | -0.7778 |

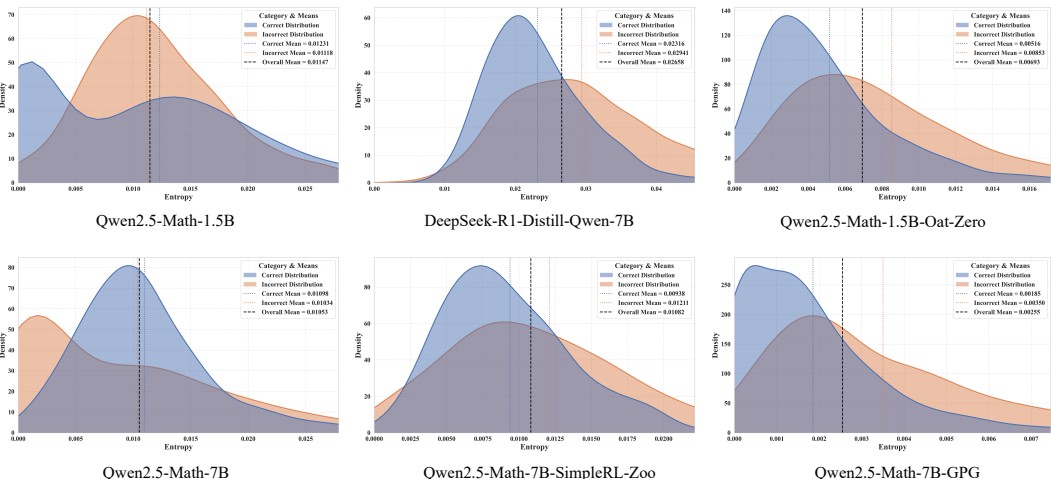

Figure 8: The entropy distribution of correct and incorrect responses within different models. The results are evaluated under the setting of temperature = 0.1.

their erroneous answers. Secondly, many correct responses exhibit high entropy, suggesting a lack of confidence even when the model produces the right answer.

This evidence highlights that relying on aggregate metrics like average entropy is insufficient, as it masks these critical sample-level discrepancies. The observed miscalibration of confidence at this granular level strongly motivates our proposed Entropy-Driven Advantage (EDA) mechanism, which is designed to apply more precise rewards and penalties to address these confidence misjudgments directly.

## A.7 USE OF LARGE LANGUAGE MODELS

Some portions of the text were polished with the assistance of Large Language Models (LLMs). All content remains the responsibility of the authors.

## A.8 ETHICS STATEMENT

This study does not involve human subjects, sensitive data, or potentially harmful applications. All mathematical reasoning datasets used (e.g., DeepScaleR, MATH, AIME) are publicly available and contain no personal privacy information or copyright-protected content. The authors have read and pledged to uphold the ICLR Code of Ethics. The research encountered no conflicts of interest, discriminatory bias, or legal-compliance risks. Every experiment followed academic integrity principles, and all reported results are truthful and unaltered.

## A.9 REPRODUCIBILITY STATEMENT

Appendix A.2 presents the complete algorithmic pseudocode, Appendix A.3 lists all hyperparameters and training schedules, and Section 5 describes the data filtering pipeline in detail. Following the prescriptions in these sections enables full replication of the reported results.

