# OpenReview forum: "EDGE-GRPO: Entropy-Driven GRPO with Guided Error Correction for Advantage Diversity"
_ICLR.cc/2026/Conference — ICLR 2026 Conference Withdrawn Submission_

### Official Review · Reviewer_eRnb · 2025-10-26

**Soundness:** 2
**Presentation:** 1
**Contribution:** 2
**Rating:** 2
**Confidence:** 4

**Summary:**

EDGE-GRPO introduces an entropy-driven variant of Group Relative Policy Optimization to mitigate advantage collapse in reasoning-oriented LLMs. It combines Guided Error Correction (GEC) that injects reference solutions to ensure response diversity and Entropy-Driven Advantage (EDA) that weights advantages by policy entropy. Empirical evaluation shows its effectiveness over multiple benchmarks and model setups.

**Strengths:**

- Some empirical findings are interesting, e.g., wrong rollout would likely to be also deterministic for model, with low entropy.
- Evaluation is validated over different mode setups, across Qwen and Llama, and with diverse benchmarks.
- The topic in RLVR this paper trying to address is timely and important.

**Weaknesses:**

- The writing clarity in format, notation, and math needs to be greatly enhanced. See my first series comment bullet points in Questions section for detail.

- Evaluation benchmark is problematic and lacks clarity. Specifically, (i) why AMC has 83 questions? the standard AMC23 benchmark used by the community only has 40 questions; (ii) The baseline performance is lower reported compared to the previous work, taking Qwen2.5-Math-7B as an example, this work reports as 53.40, while previous published work reports 64.3 for the same baseline (in COLM 2025, arxiv 2504.07086), which raises concerns regarding evaluation protocols; (iii) for AIME, AMC these benchmark with less questions, reporting pass@1 is not enough, community common practice would at least report avg@8 or avg@16.

- Training configuration is also unclear and doesn't sound. (i) All problems in DeepScaleR should be verifiable. In standard VeRL training, including a default Qwen training prompt to instruct the model to put its final answer into box. Also, the verifier would also extract the final number if the model failed to put its answer into the box. Therefore, I don't consider the rationale for truncating the dataset sounds. The number of training source and epoch of training is inadequate; (ii) Why the macro-batch is only 1, i.e., you have 1000 questions and you only trained for 1000 steps, which means that in each macro-batch step, only 1 prompt is sampled. I am highly concerned with, as this definitely doesn't align with the community practice, which usually have the macro-batch of 64/128. This non-standard practice would affect the other baseline performance you compared with. Thus, this batch size is unusually small, and the variance in policy gradients could bias the results or make your result less reproducible but with only single-run result reported.

- The ablation is not robust. I cannot see which technique contributes more, as their effects appear in some randomness.

**Questions:**

Following from Weakness 1, following trivial mistakes must be fixed in the next version:
- Specifically, writing and formatting hinders readability, please properly use \citep or \citet in the revised version.
- Math notation also hinders clarity. Please don't make both policy entropy and logit probability from $\pi$ both with $P$ (e.g., eq3).
- I am concerned that in eq4, softmax is divided by logit by $T$? Why to rescale it, what you simply meant is that response with longer one would naturally have lower entropy.
- How to calculate entropy_{correct} and entropy_{wrong}, please clarify.
- You are now also using some operation probabilities under $P$, e.g., $P_{1}$. Notationally, it reads really confusing, $P$ represents both entropy, logit probability, and guided operation probability in your paper. Please fix all these to improve the readability.
- $P_{1}=\frac{2}{P}$ doesn't ensure at least one response will be replaced if I understand your definition correctly. $P(\text{at least one}) = 1 - \left(1 - \frac{2}{G}\right)^{G} \approx 1 - e^{-2} \approx 0.865$ according to my calculation, please carefully reframe your claim to ensure math rigor.

The claims from line 200-202 versus line 203-209 reads very unclear. The upper para is saying that self-reflection could improve the accuracy, but the next paragraph zoom into "forced reflection" and saying that this cannot improve model performance, what's the difference here? Please clarify.

It would better to evaluate the percentage of all-negative-reward group, as your aim to target advantage collapse.

---

### Official Review · Reviewer_CiZB · 2025-10-31

**Soundness:** 3
**Presentation:** 3
**Contribution:** 2
**Rating:** 4
**Confidence:** 4

**Summary:**

The paper proposes EDGE-GRPO, an algorithm intended to mitigate the advantage collapse problem in GRPO-based RL for mathematical reasoning. It introduces two modules:
(1) Guided Error Correction (GEC), which replaces or injects reference solutions or regeneration prompts to ensure diverse correct/incorrect responses;
(2) Entropy-Driven Advantage (EDA), which multiplies the normalized advantage by normalized entropy to encourage “confidence-aligned” advantage shaping.

**Strengths:**

1. Focus on an important question on the GRPO advantage collapse.

2. Contains extensive empirical evaluation across multiple benchmarks.

3. High practical relevance due to increasing community interest in small-data RL.

**Weaknesses:**

1. 1The motivation for EDA relies on Figure 3, which claims that models often assign low entropy (high confidence) to incorrect responses. However, the paper computes entropy comparisons globally across all responses, not within each question, which is the only setting relevant for GRPO’s intra-group advantage ranking. Prior work [1] shows that LLMs generally assign higher confidence to the correct answer within each question. Therefore, a global statistic cannot support the claim that GRPO’s per-question ordering is unreliable. To justify EDA, the authors should measure whether incorrect responses have lower entropy than correct ones within the same question, not relative to a global average. As written, the analysis in Figure 3 does not meaningfully support the algorithm’s motivation.

2. The paper concludes that reflection is ineffective based on single-pass forced reflection. However, prior work [2] shows that increasing the reflection/thinking budget can raise accuracy from <20% to ~60%. This indicates that reflection ability is limited only under low compute budgets, not inherently weak. Since the paper evaluates reflection with just one reflection step, the conclusion that “reflection is ineffective” is not well supported.

3. One straightforward way to utilize negative signals in all-incorrect groups is NSR (Negative Sample Reinforcement), which just penalizes them. NSR already tackles the same issue, extracting learning signals from incorrect responses, without requiring external reference solutions. Since the paper does not compare against applying NSR in the all-incorrect setting, it is unclear whether the proposed method provides any benefit beyond this existing negative-sample method.

4. The algorithm assumes the training question has a high-quality chain-of-thought reference solution, which is often the main bottleneck in practice.

[1] Probabilities of Chat LLMs Are Miscalibrated but Still Predict Correctness on Multiple-Choice Q&A

[2] S1: Simple Test-Time Scaling

[3] The Unreasonable Effectiveness of Entropy Minimization in LLM Reasoning

**Questions:**

See weakness

---

### Official Review · Reviewer_7ys9 · 2025-10-31

**Soundness:** 2
**Presentation:** 2
**Contribution:** 2
**Rating:** 2
**Confidence:** 5

**Summary:**

This paper proposes EDGE-GRPO, a RL framework that mitigates the advantage collapse problem in GRPO-based reasoning models. The method integrates Entropy-Driven Advantage (EDA) to enhance training signal diversity and Guided Error Correction (GEC) to improve response diversity. In addition, the authors comprehensive analyze self-reflection behaviors and rollout entropy in LLM reasoning. Experiments across multiple reasoning benchmarks show that EDGE-GRPO achieves strong performance with minimal training data, outperforming vanilla GRPO and competing baselines.

**Strengths:**

1. The authors conduct a comprehensive study on the helpfulness of self-reflection and the role of entropy in both correct and incorrect responses within LLM reasoning tasks.
2. The paper proposes two strategies to process cases where all responses are incorrect during RL rollouts and introduce an entropy-based augmentation to better differentiate advantages in a group.

**Weaknesses:**

The motivation behind the proposed method is sound; however, the experimental settings and design exhibit several deficiencies.

1. Unreasonable experimental setup: The experimental settings are not sufficiently rigorous. Training an RLVR model with only 1K problems for a single epoch is far from convergence, as evidenced by prior works [1,2]. Moreover, the authors train their model on a challenging dataset but restrict the maximum response length to 1,024 tokens, which may significantly limit performance compared with established practices [3].

2. Unreasonable evaluation setup: The AMC and AIME benchmarks contain relatively few problems—for instance, one correct answer in AIME corresponds to a 3.3-point change in accuracy. Without multiple sampling runs, such evaluations can exhibit significant instability and variance. Moreover, EDGE-GRPO incorporates a Reference Solution (the source of which is not clearly specified by the authors), making the comparison in Table 1 potentially unfair. Alternatively, Table 1 should include other methods that also leverage reference solutions, such as Luffy [4], for a more equitable comparison.

[1] Dapo: An open-source llm reinforcement learning system at scale.

[2] Prorl: Prolonged reinforcement learning expands reasoning boundaries in large language models.

[3] SimpleRL-Zoo: Investigating and Taming Zero Reinforcement Learning for Open Base Models in the Wild

[4] Learning to Reason under Off-Policy Guidance

**Questions:**

1. In line 192, the authors state that responses from models without distillation containing self-reflection often exhibit lower accuracy. Could this be because only more challenging problems tend to elicit self-reflection, and the incorrect responses arise from the intrinsic difficulty of those problems rather than the self-reflection itself?

2. Regarding the Direct Answer Injection strategy, is it possible that the model might produce responses with correct final answers but incorrect reasoning? If the rollout accuracy is near zero, this could indicate that the task is too challenging for the policy. In such cases, even appending the correct final answer to the prompt may fail to guide the model toward valid intermediate reasoning, leading it instead to generate seemingly plausible yet incorrect rationales.

---

### Official Review · Reviewer_Q2Ng · 2025-10-31

**Soundness:** 2
**Presentation:** 3
**Contribution:** 2
**Rating:** 4
**Confidence:** 4

**Summary:**

The paper proposed a reinforcement learning algorithm called Edge-GRPO for post-training mathematical reasoning models. The author proposes the Guided Error Correction Module, which aims to resolve the hard question beyond the model's capability as well as the Entropy-Driven Advantage, which incorporates the relative entropy as reward signals.

**Strengths:**

The illustration and writing of the method are clear and easy to follow.

**Weaknesses:**

Major Weakness:
* I’m doubtful about the fairness of the experimental setup. Since reference solutions are used, the proposed method benefits from additional distilled knowledge compared to other reinforcement learning methods such as Dr.GRPO, DAPO, and vanilla GRPO. I believe a fairer baseline would be SFT+GRPO. From Tables 1 and 2, it appears that the proposed method without reference solutions performs more or less on par with the other RL methods.

Minor Weakness:
* In terms of writing, I found the analyses in Figures 2 and 3 somewhat disconnected from the methodology. The three points shown in Figure 3 are either obvious or already well-studied in prior work. Section 3.1 essentially states that reflection is limited in extending the model’s capability boundaries, so it is not used. For Figure 3, it’s unclear how it inspires the use of entropy in the rewards. Given that Section 3 is titled “Investigation of Advantage Collapse in GRPO,” I expected to see patterns in training dynamics related to GRPO.
* In Figure 5, lower advantage variance does not necessarily indicate advantage collapse, nor does it directly reflect training dynamics or model quality. Ultimately, what matters—at least in the context of this paper—is whether the model is confident in correct responses and uncertain in incorrect ones. Such an analysis is missing from the paper.
* The use of entropy in the objective is not new. This is not a major issue, as many concurrent works explore similar ideas. However, I would have liked to see how entropy reweighting performs in the rewards directly—for example, by directly replacing the GRPO/Dr.GRPO advantage functions.

**Questions:**

See above

---

### Note · Authors · 2025-11-14

**Comment:**

We sincerely appreciate all the questions and suggestions raised by the reviewers!

We agree with the reviewers on the potential improvements in the experimental setup, including the batch size, maximum generation length, and the scale of the training data. However, we must clarify that the current settings were strictly limited by our computational resources; we only had eight A100-40G GPUs available for the experiments. And we will also improve the evaluation regarding AIME and AMC.

Regarding the reviewers' discussion about the performance improvements brought by reflection, we have already discussed some relevant paper in the second paragraph of the "Related Work" section, pointing out that there is currently no unified answer as to whether reflection is truly effective. Therefore, we conducted a relatively comprehensive set of experiments, selecting models of different sizes to analyze the accuracy of both spontaneous reflection and forced reflection. When forcing the model to reflect, we also experimented with multiple reflection prompts.

Additionally, we will continue to improve aspects of the paper such as the symbolic notation and the ablation studies. Considering all the points mentioned above, we have decided to withdraw the submission for further refinement and revision.

**Withdrawal Confirmation:**

I have read and agree with the venue's withdrawal policy on behalf of myself and my co-authors.